# Direct 3D-Aware Object Insertion via Decomposed Visual Proxies

**Jingbo Gong** [1 3]  **Yikai Wang** [2]  **Yushi Lan** [2]  **Yuhao Wan** [1]  **Ziheng Ouyang** [1]
**Rui Zhao** [4]  **Ming-Ming Cheng** [1 5]  **Qibin Hou** [1]  **Chen Change Loy** [2]

Project page: https://gong1130.github.io/DIRECT

## Abstract

Object insertion aims to seamlessly composite a reference object into a specified region of a background image. Recent diffusion-based methods achieve high visual quality but formulate insertion as a simple 2D inpainting task, providing no explicit control over the object's 3D pose and limiting their practical applicability. We propose **DIRECT** (**D**ecomposed **I**njection for **RE**ference **C**omposition and **T**arget-integration), a novel framework that integrates interactive pose manipulation with high-fidelity 2D image synthesis to enable pose-controllable object insertion. Our method decomposes the insertion conditions into three complementary components: appearance guidance capturing visual details from the reference object, geometry guidance derived from the user-adjusted 3D proxy, and context guidance from the target background. By injecting them through separate pathways, DIRECT avoids feature entanglement and simultaneously preserves reference appearance, follows the user-specified pose, and adapts the object to the target scene. We also introduce an automated data construction pipeline to improve the diversity and quality of training data. Experiments show that DIRECT outperforms previous methods in both geometric controllability and visual quality.

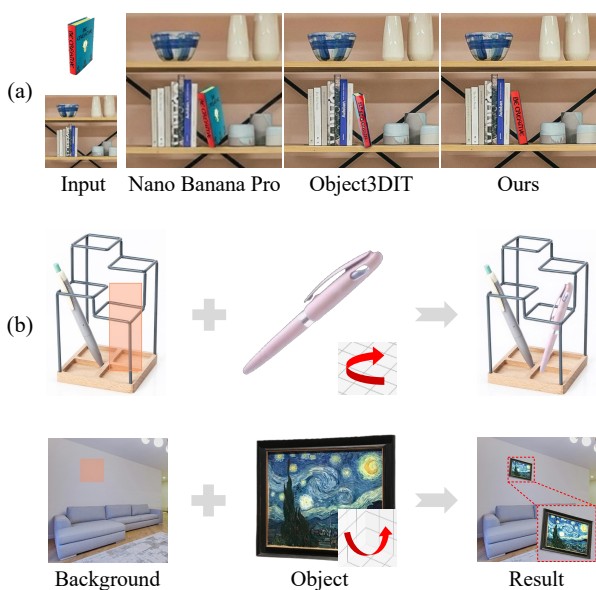

*Figure 1.* **Pose-controllable object insertion.** (a) Existing pipelines have difficulty placing the reference object in a reasonable and user-specified pose within the background image, even when using a strong 2D generative model such as Nano Banana Pro (Google, 2025) or a 3D-aware editing model such as Object3DIT (Michel et al., 2023). In contrast, our framework inserts the object with precise pose control and better scene alignment. (b) Additional results show that our method achieves high-fidelity insertion with precise pose control in complex real-world scenes. We report the prompts used for competing methods in Appendix A.

## 1. Introduction

Object insertion has recently advanced significantly through reference-guided image generation (Chen et al., 2024; Song et al., 2026). Leveraging powerful generative backbones like Stable Diffusion (Rombach et al., 2022) and FLUX (Black Forest Labs, 2024), these methods achieve impressive fidelity in identity preservation and environmental harmonization. However, they are confined to the 2D image plane, lacking the capability to explicitly control the object's 3D pose. This deficiency restricts their applicability in practical scenarios where precise spatial alignment is essential. To address this, we investigate the problem of *Pose-Controllable Object Insertion*. This task imposes a rigorous geometric

---

[1]VCIP, School of Computer Science, Nankai University [2]S-Lab, Nanyang Technological University [3]Zhongguancun Academy [4]SenseTime Research [5]NKIARI, Shenzhen Futian. Correspondence to: Ming-Ming Cheng <cmm@nankai.edu.cn>, Yikai Wang <yi-kai.wang@outlook.com>.

Work done while Jingbo Gong was visiting S-Lab, Nanyang Technological University. Yikai Wang is now at Meta. Yushi Lan is now at the University of Oxford.

*Proceedings of the 43rd International Conference on Machine Learning*, Seoul, South Korea. PMLR 306, 2026. Copyright 2026 by the author(s).

constraint beyond conventional appearance consistency: the synthesis must be guided by a specified 3D pose rather than solely by 2D appearance context.

As illustrated in Fig. 1(a), current approaches generally struggle to meet this rigorous requirement. This is primarily due to the inherent limitations of their control mechanisms. **(1)** Text-guided models, such as Nano Banana Pro (Google, 2025), rely on natural language, yet language is spatially ambiguous. For instance, abstract descriptions like "leaning against" are often under-defined, failing to specify the exact contact geometry. This frequently leads the model to hallucinate plausible but incorrect poses to fit the semantic context. **(2)** Parametric 3D-aware models, like Object3DIT (Michel et al., 2023), attempt to inject explicit control via rotation angles. However, establishing a precise mapping from these abstract scalars to dense pixel-level deformations presents a significant challenge. Lacking explicit spatial correspondence, the model struggles to translate low-dimensional parameters into the correct geometric projection.

To overcome these hurdles, we propose DIRECT, a generative framework that integrates explicit 3D visual proxies to enable precise Pose-Controllable Object Insertion. Instead of relying on sparse or ambiguous control signals, we leverage recent feed-forward image-to-3D models (Xiang et al., 2025) to lift the reference image into a coarse 3D representation. This proxy is then rendered under the specified 6-DoF pose to yield a dense geometric condition image. Guided by this explicit condition, our framework bridges the representational gap, ensuring the inserted object strictly adheres to the target pose (as demonstrated in Fig. 1(b)).

However, the rendered 3D proxy often suffers from texture degradation and visual artifacts compared to the original image. Consequently, naively conditioning the generator on this proxy can introduce noise or even confuse the generation process. To address this challenge and explicitly model the available conditioning signals, we propose to decompose the input conditions for this task into three complementary components. Specifically, we separate the guidance into orthogonal sources: geometry from the 3D proxy, appearance from the reference object, and context from the target scene. By injecting these signals through independent pathways, our framework allows the model to strictly adhere to geometric constraints while simultaneously leveraging high-fidelity textures and environmental lighting cues, enabling realistic object-scene synthesis.

Training such models requires large-scale paired data capturing complex real-world scenes, yet existing object-centric 3D datasets (Reizenstein et al., 2021; Yu et al., 2023) are limited to simplified environments with low visual fidelity, restricting object diversity and natural background coherence. To overcome this, we propose an automated pipeline that synthesizes paired training samples from single-view, in-the-wild images. Our approach filters high-quality object instances using a VLM (Bai et al., 2025)-powered agent and generates novel views via a generative editing model (Wu et al., 2025), preserving visual quality while introducing diverse scenes and rich background interactions. Using this pipeline, we curate a hybrid dataset of over 160k pairs by combining synthesized samples from SA-1B (Kirillov et al., 2023) with a high-quality subset of MVImgNet (Yu et al., 2023), substantially improving model generalization in real-world applications.

To validate DIRECT, we conduct extensive evaluations on the testing split of our curated hybrid dataset. The results demonstrate that our method consistently outperforms baselines, achieving superior scores in both reconstruction quality and identity preservation. Notably, our approach exhibits remarkable robustness to artifacts in upstream 3D priors and accurately handles complex pose transformations, effectively addressing common issues such as geometric distortion and texture degradation in existing methods.

In summary, our contributions are threefold:

- We present DIRECT, a generative framework that leverages explicit 3D geometric conditions to bridge the gap between rigid 3D control and high-fidelity 2D synthesis. By converting 6-DoF pose requirements into dense geometric conditions, we enable precise object insertion without relying on ambiguous text or sparse parameters.
- We propose to decompose the guidance into three complementary components (appearance, geometry, context) and inject them through independent pathways, helping the model better balance these signals in synthesis.
- We introduce an automated data construction pipeline. By synthesizing reference views from single-view real-world images to construct training pairs, we significantly expand the dataset diversity in complex real-world scenes and thus improve model generalization.

## 2. Related Work

***Object insertion*** has evolved from semantics-driven synthesis (Yang et al., 2023; Song et al., 2023) to identity preservation. IMPRINT (Song et al., 2024) and AnyDoor (Chen et al., 2024) enhance fidelity through feature injection. SEELE (Wang et al., 2024) adopts a copy-paste-harmonize workflow. InsertAnything (Song et al., 2026) utilizes the FLUX (Black Forest Labs, 2024) backbone and a "diptych" design (Cao et al., 2024) to reformulate object insertion as a unified inpainting task. However, these methods generally operate on the 2D image plane. They lack explicit geometric controllability, typically failing to handle scenarios that require precise, user-defined 3D pose manipulation.

***3D-aware image editing*** methods generally fall into three paradigms. Object3DIT (Michel et al., 2023) and Neural

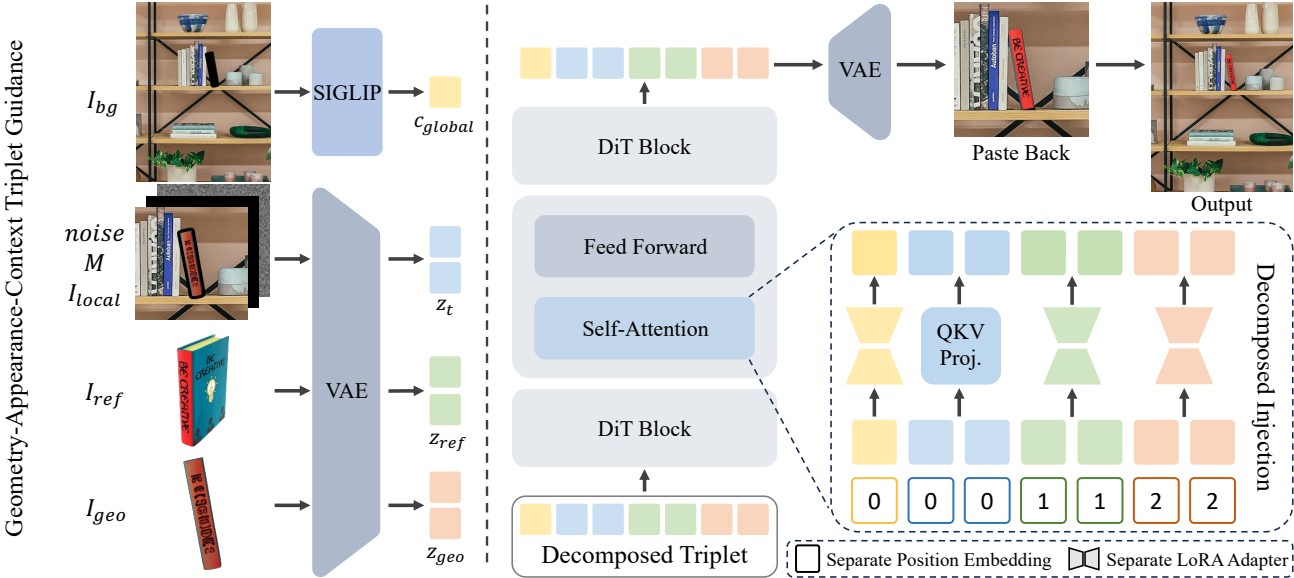

*Figure 2.* **Illustration of our framework.** The generation process is controlled by three types of conditions: appearance guidance from the original reference object, geometry guidance from the rendered image with the user-specified pose, and context guidance from global features of the background image. These conditions are injected through decomposed LoRA pathways to reduce interference. The standard masked background condition is modified by pasting the rendered object with the desired pose into the masked region. The editing region is cropped for focused local insertion and pasted back into the high-resolution image after generation.

Assets (Wu et al., 2024) fine-tune generative models with encoded geometric signals, such as camera parameters or bounding boxes. However, these abstract controls create a "cognitive gap" and struggle to align fine-grained details with geometry. Training-free methods, such as Diffusion Handles (Pandey et al., 2024), GeoDiffuser (Sajnani et al., 2025), and Image Sculpting (Yenphraphai et al., 2024), manipulate diffusion features via inversion, but suffer from high test-time optimization costs. 3D asset-based methods, such as ZeroComp (Zhang et al., 2025) and 3D Copy-Paste (Ge et al., 2023), leverage intrinsic 3D cues but require high-quality 3D assets, which are difficult to obtain from single-view images. Closest to our work are methods that use geometric proxies as guidance (Ge et al., 2023; Liu et al., 2025). However, they are fundamentally designed for in-place editing and lack the ability to perform insertion.

In contrast, our framework lifts a single image into a visual 3D proxy as an explicit control signal. By injecting the rendered proxy, reference image, and target scene context into the generative model through decomposed pathways, we achieve precise pose control, high-fidelity identity preservation, and realistic scene integration without requiring high-quality 3D assets or test-time optimization.

***Image-to-3D generation*** has undergone a significant paradigm shift, transitioning from computationally intensive per-scene optimization toward efficient, feed-forward inference. Early approaches, represented by DreamFusion (Poole et al., 2023) and Magic3D (Lin et al., 2023), leveraged Score

Distillation Sampling (SDS) to optimize NeRF (Mildenhall et al., 2021) representations, capable of generating 3D assets but suffering from slow per-object optimization. To address the efficiency bottleneck, feed-forward approaches like LRM (Hong et al., 2024) and LGM (Tang et al., 2024) emerged, utilizing transformer-based architectures to directly regress 3D representations from a single image in seconds. Most recently, 3D diffusion models such as GaussianAnything (Lan et al., 2025), TRELLIS (Xiang et al., 2025), and Hunyuan3D (Lai et al., 2025) have set new standards for geometric topology and texture fidelity. In our work, we leverage these advancements to employ a 3D proxy as an interactive geometric condition, bridging explicit 3D pose control and flexible 2D image generation.

## 3. Method

### 3.1. Pose-Controllable Object Insertion

We formalize the task of pose-controllable object insertion as a conditional image generation problem. Let $\mathcal{I} \subseteq \mathbb{R}^{H \times W \times 3}$ denote the image space. We are provided with a reference object image $I_{ref} \in \mathcal{I}$, a target background image $I_{bg} \in \mathcal{I}$, and a binary mask $M \in \{0,1\}^{H \times W}$ indicating the insertion region. Unlike standard subject-driven inpainting, which seeks to maximize the likelihood $p(I_{out} \mid I_{ref}, I_{bg}, M)$ based solely on semantic compatibility, our problem imposes a strict geometric constraint. The inserted object must conform to a user-specified 6-DoF pose $\boldsymbol{\xi} \in \mathfrak{se}(3)$ relative to the background scene.

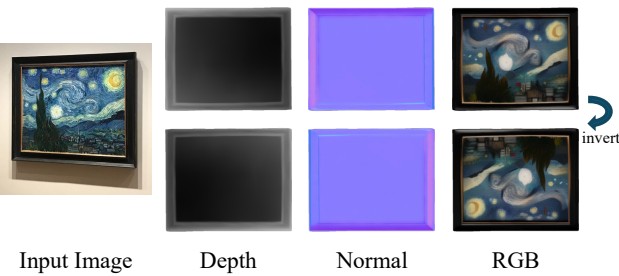

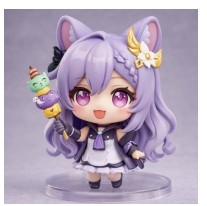
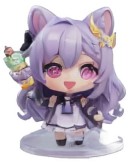
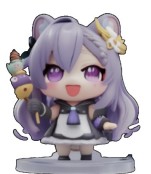

Input Image     Depth     Normal     RGB

*Figure 3.* **Geometric semantic ambiguity.** Standard spatial signals, such as depth and normal maps, fail to distinguish the orientation of symmetric objects, whereas our RGB geometric condition explicitly preserves semantic pose.

Input Image        LGM        TRELLIS

*Figure 4.* **Appearance fidelity gap.** Current image-to-3D models suffer from severe texture degradation. Relying solely on the rendered proxy can lead to blurry outputs, motivating the re-injection of the original reference.

***3D Visual Proxy Lifting.*** The reference object image is 2D, while user interaction is more intuitive when the object can be directly translated and rotated in 3D space. In contrast, standard 2D diffusion models lack an intrinsic understanding of $\mathfrak{se}(3)$ transformations. We bridge this modality gap by lifting the 2D reference object $I_{ref}$ into a manipulable 3D proxy $\mathcal{P}$. Users interact with this proxy to specify the desired pose $\boldsymbol{\xi}$, which is then rendered as a dense geometry guidance image $I_{geo}$. We provide our implementation of this user-friendly interaction in Sec. C.

***Decomposed Generative Objective.*** While $I_{geo}$ provides precise geometry guidance, it often suffers from texture degradation due to the limitations of single-view 3D reconstruction. Conversely, $I_{ref}$ contains high-fidelity texture but lacks the desired spatial arrangement. To reconcile these complementary signals, we formulate the generation of the output image $I_{out}$ as learning the distribution $p_\theta$ conditioned on a decomposed set of guidance signals:

$$I_{out} \sim p_\theta(I \mid \underbrace{I_{ref}}_{\text{Appearance}}, \underbrace{I_{geo}}_{\text{Geometry}}, \underbrace{\Psi(I_{bg})}_{\text{Context}}, M), \quad (1)$$

where $\Psi(\cdot)$ represents the global context encoding that provides scene-level semantics. Our objective is to optimize the parameters $\theta$ such that $I_{out}$ inherits high-frequency identity details from $I_{ref}$, strictly adheres to the pose defined by $I_{geo}$, and harmonizes photometrically with $I_{bg}$.

## 3.2. Geometry-Appearance-Context Triplet Guidance

Figure 2 illustrates the overall framework. To achieve precise object insertion, the generative model must satisfy three distinct and often conflicting requirements: it must adhere strictly to the user-defined pose (Geometry), preserve the identity of the reference (Appearance), and harmonize with the background (Context). We propose to decompose the conditioning signal into a *visual triplet*, handling each component through a specialized pathway.

The **Geometry** guidance is provided by the rendered dense geometry image $I_{geo}$. Standard geometric signals are often semantically ambiguous. As shown in Fig. 3, the depth map and normal map of a symmetric object, such as a painting, look identical whether upright or upside-down. Our RGB-based geometry guidance ($I_{geo}$) removes this ambiguity, ensuring the model orients the object correctly.

The **Appearance** guidance, $I_{ref}$, provides reliable identity information about the reference object. While the 3D proxy provides explicit pose guidance, its rendered texture is often blurry or degraded (Fig. 4). Relying solely on geometry guidance would degrade the output quality. Therefore, we re-inject the original reference image $I_{ref}$ to recover high-fidelity appearance details.

The **Context** guidance enables high-resolution insertion while preserving global scene awareness. A major challenge in object insertion is the trade-off between resolution and context: cropping the image focuses on the object but loses the surrounding environment (lighting sources, perspective lines). We resolve this by processing the background at two levels. Locally, the generator operates on a high-resolution crop around the mask region. We form a local composite input $I_{local}$ by pasting $I_{geo}$ into $I_{bg}$ within $M$, and feed the cropped pair $(I_{local}, M)$ to the inpainting backbone. Globally, we encode the full-frame background $I_{bg}$ to obtain global semantic tokens $c_{global}$. This allows the model to attend to the entire scene's lighting and composition through the attention layers, ensuring the locally inserted object harmonizes with the global environment.

***Decomposed Triplet Injection.*** Merging these signals is non-trivial. With naive concatenation and LoRA fine-tuning, as in prior works such as Tan et al. (2025) and Ouyang et al. (2025), the model often exhibits condition interference. It over-relies on the geometry proxy, producing outputs that follow $I_{geo}$ in pose while inheriting its degraded appearance and ignoring the high-fidelity reference $I_{ref}$. Examples of this degradation are provided in Sec. 4.4 (see Fig. 9). To solve this, we employ a *Decomposed Injection Strategy*.

Both the reference image $I_{ref}$ and the geometric condition $I_{geo}$ are encoded into latent tokens $z_{ref}$ and $z_{geo}$. The noisy target latent at timestep $t$ is denoted as $z_t$. The background image $I_{bg}$ is encoded by a frozen SIGLIP (Tschannen et al.,

2025) encoder into global context tokens $c_{global}$. To distinguish these condition tokens, we employ two mechanisms: (1) Independent Positional Embedding: We assign distinct Rotary Positional Embeddings (RoPE) (Su et al., 2024) to the appearance and geometry tokens, spatially isolating them in attention. The global context tokens encode scene-level semantics rather than pixel-aligned spatial structure, so they are not assigned a separate spatial positional encoding. (2) Modality-Specific Adapters: We introduce separate LoRA (Hu et al., 2022) adapters for each condition within the self-attention layers. This forces the model to learn condition-specific transformations: one extracts structural pose information from $z_{geo}$, one extracts identity and texture from $z_{ref}$, and one extracts global context from $c_{global}$.

In summary, our model processes a unified token sequence $Z = [c_{global}, z_t, z_{ref}, z_{geo}]$ through these decomposed pathways, synthesizing results that satisfy the full Geometry-Appearance-Context triplet.

### 3.3. Data Construction

***Limitations of Existing Datasets.*** To train our model for precise pose control, we require training pairs consisting of an isolated reference object image $I_{ref}$ and a target image $I_{gt}$ depicting the same object instance in a distinct pose within a background. A conventional strategy is to utilize object-centric 3D datasets, such as MVImgNet (Yu et al., 2023), which provide videos of objects captured from multiple angles. However, relying solely on such datasets introduces three critical limitations that hinder generalization to in-the-wild scenarios: **(1)** Simplistic backgrounds: Objects are often captured in clean or empty environments, lacking the realistic interactions between the inserted object and the surrounding elements. **(2)** Restricted viewpoints: Camera trajectories are typically orbital top-down views, lacking diversity in elevation and perspective. **(3)** Image quality artifacts: Images extracted from videos frequently suffer from motion blur, degrading the generation quality.

To overcome these limitations, we propose an automated pipeline to curate high-quality training pairs from single-view in-the-wild images. This pipeline operates in two sequential stages: First, an intelligent agent filters high-quality object instances. Second, a generative editing model synthesizes novel views to form paired references.

***Step 1: Automated object curation via VLM agent.*** We construct an intelligent agent leveraging Qwen3-VL (Bai et al., 2025) and SAM-3 (Carion et al., 2025) to identify and filter suitable objects with precise masks based on saliency, structural completeness, and segmentation precision. The process operates in three steps:

- Propose: The agent analyzes the high-resolution image to propose categories of salient objects present in the scene.

- Segment: Guided by these proposed categories, SAM-3 generates candidate masks for all corresponding instances.
- Verify: The agent acts as a judge to discard occluded or truncated objects. It performs a "zoom-in check", examining a localized crop around the mask to verify both the structural completeness of the object and the precision of the mask boundaries.

With this agent, we filter images containing fully visible, high-fidelity, diverse objects within complex scenes.

***Step 2: Synthesizing references via view transformation.*** To construct training pairs from these single-view images, we employ a "Real-Target, Synthetic-Source" strategy. The original real-world image is treated as the ground truth target $I_{gt}$, while the input reference image $I_{ref}$ is synthesized by a generative model. The object is first extracted using the verified mask. Then, Qwen-Image-Edit (Wu et al., 2025), equipped with an angle-editing adapter[1], is utilized to rotate the object to a random novel view, which serves as the reference input $I_{ref}$. By anchoring the ground truth to real-world captures, our constructed dataset features complex background compositions, diverse viewpoints, and high image quality, effectively overcoming the limitations of standard object-centric 3D datasets. Note that although Qwen-Image-Edit maintains object identity well, it only provides approximate viewpoint changes and does not perform object insertion. We therefore use it only to synthesize appearance-preserving reference views for data construction, rather than as a solution to our pose-controllable object insertion task.

We processed a subset of SA-1B (Kirillov et al., 2023) to construct 65k high-quality pairs and combined them with 93k filtered MVImgNet (Yu et al., 2023) samples, yielding a hybrid dataset of approximately 160k pairs that balances real-world scene complexity with 3D consistency. Further details on training data construction, including object curation and quality filtering, are provided in Appendix B.

### 3.4. Training Strategy

We freeze the backbone and train only the LoRA adapters and linear projectors using the standard rectified flow matching objective (Lipman et al., 2023; Esser et al., 2024).

***Shape-Decomposed Mask Augmentation.*** A naive training approach utilizes the ground truth mask of the target object in $I_{gt}$ as the inpainting mask $M$. However, we observe that the model tends to overfit to the precise mask boundary. This leads to "shape leakage", where the model ignores the geometric condition $I_{geo}$ and simply fills the exact contour provided by the mask. To mitigate this, we employ a shape-decomposed mask augmentation strategy to decompose the inpainting region from the object's actual geometry. During

---

[1]https://huggingface.co/dx8152/Qwen-Edit-2509-Multiple-angles

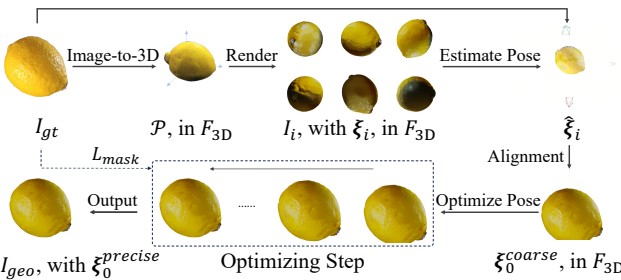

*Figure 5.* **Overview of geometric alignment pipeline.** Given a target image $I_{gt}$, we estimate the rendering pose of the 3D proxy $\mathcal{P}$ such that its projection matches the target object. The pose-aligned rendering is then used as the geometric condition $I_{geo}$ for training.

training, the precise mask is replaced with a random real-object mask sampled from an external dataset (Wang et al., 2025b). This prevents the model from treating the mask boundary as a shortcut, encouraging it to rely more on the intended geometry and appearance conditions rather than the inpainting mask.

***Progressive Resolution Training.*** To balance training efficiency with high-quality generation, we adopt a two-stage progressive training strategy applied to the local processing window. In the first stage, we train with fixed $512^2$ crops. This phase allows the model to efficiently learn the fundamental capability. Subsequently, we fine-tune the model using arbitrary aspect ratios with approximately $1024^2$ pixels to achieve high-resolution synthesis while naturally accommodating diverse object geometries.

***Geometric Alignment and Proxy Rendering.*** In inference, the target pose $\boldsymbol{\xi}$ is specified by the user. However, during training, we must automatically derive the geometric condition $I_{geo}$ that aligns with the ground truth target image $I_{gt}$. To achieve this, we use a pose estimation pipeline to determine the optimal 6-DoF parameters for the 3D proxy $\mathcal{P}$ such that its projection matches the object in $I_{gt}$.

The alignment is treated as an offline pre-processing step, and its overall workflow is illustrated in Fig. 5. Given a target image $I_{gt}$, we first reconstruct a 3D representation $\mathcal{P}$ using an image-to-3D model (Xiang et al., 2025), which defines a canonical coordinate system $F_{3D}$. In $F_{3D}$, we render $\mathcal{P}$ under six predefined camera poses $\{\boldsymbol{\xi}_i\}_{i=1}^{6}$ (front/back/left/right/up/down) to obtain six images $\{\tilde{I}_i\}_{i=1}^{6}$. Then, we feed the seven-image set $\{\tilde{I}_1, \ldots, \tilde{I}_6, I_{gt}\}$ into VGGT (Wang et al., 2025a) to estimate relative poses $\{\hat{\boldsymbol{\xi}}_k\}_{k=0}^{6}$, where $\hat{\boldsymbol{\xi}}_0$ corresponds to $I_{gt}$ and $\hat{\boldsymbol{\xi}}_i$ corresponds to $\tilde{I}_i$. Since $\{\hat{\boldsymbol{\xi}}_k\}$ is defined up to an unknown global similarity transform, we recover $S \in \mathrm{Sim}(3)$ by aligning the six anchor views. Applying $S$, we map the target camera pose from the VGGT coordinate system into the proxy coordinate system $F_{3D}$, yielding the coarse estimate $\boldsymbol{\xi}_0^{\mathrm{coarse}}$. Starting from $\boldsymbol{\xi}_0^{\mathrm{coarse}}$, we further refine the pose with a differentiable 3D Gaussian Splatting renderer. The camera pose parame-

ters $\phi$ are treated as learnable variables, and we optimize a silhouette consistency loss between the rendered alpha matte $\alpha_{\phi}$ and the target mask $M$: $L_{\mathrm{mask}}(\phi) = \|\alpha_{\phi} - M\|_1$. This yields the final refined pose $\boldsymbol{\xi}_0^{\mathrm{precise}}$, which is used to render and cache $I_{geo}$ for efficient training.

# 4. Experiments

***Implementation Details.*** We implement our generator based on the pre-trained FLUX.1-Fill-dev (Black Forest Labs, 2024) model. The rank of the LoRA (Hu et al., 2022) adapters is set to $128$. To enable classifier-free guidance (CFG) (Ho & Salimans, 2021) for appearance control, we randomly drop the reference condition with a probability of $0.1$ during training. Training is performed on our curated hybrid dataset using the AdamW optimizer with $\beta_1 = 0.9$, $\beta_2 = 0.999$, and a learning rate of $1 \times 10^{-4}$. The first stage trains for 200k steps on 4 A100 GPUs with a batch size of 4. The second stage trains for 40k steps on 8 A100 GPUs with a batch size of 8. During inference, the Euler scheduler is used with 28 sampling steps and the CFG scale is set to 2.0.

***Baselines.*** Since most 3D-aware editing approaches are limited to global image manipulation and cannot directly perform object insertion, we construct composite baselines by cascading 3D pose-editing tools with strong 2D insertion models. Specifically, Object3DIT (Michel et al., 2023) serves as the representative 3D-aware editing model, while TRELLIS (Xiang et al., 2025) serves as the image-to-3D model for target-pose rendering. As Object3DIT is built upon Stable Diffusion v1 (Rombach et al., 2022), we categorize these baselines into two groups for fair comparison: **(1)** Stable Diffusion-based category: Object3DIT and TRELLIS are combined with AnyDoor (Chen et al., 2024), an SD-based inserter, and compared against our SD-based variant; **(2)** FLUX-based category: Object3DIT and TRELLIS are combined with InsertAnything (Song et al., 2026), a FLUX.1-based inserter, and compared against our final FLUX-based model. Appendix D further evaluates an additional intrinsic-guided compositing baseline.

***Evaluation Benchmarks.*** We randomly sample 200 image pairs from our hybrid dataset to construct the evaluation benchmark, ensuring no overlap with the training set. The dataset consists of two subsets: 100 pairs from MVImgNet (Yu et al., 2023) representing real-world observations, and 100 pairs derived from SA-1B (Kirillov et al., 2023) via our automated pipeline. Notably, we manually verify the SA-1B subset to ensure consistency between the synthesized reference and the ground truth target, strictly excluding any samples with generation artifacts.

***Evaluation Metrics.*** We evaluate the generated results using six metrics to assess image fidelity, identity preservation, and pose accuracy. To measure reconstruction qual-

*Table 1.* **Quantitative comparison.** Our framework consistently achieves the best results across all metrics under different backbones. For Object3DIT (Michel et al., 2023) and TRELLIS (Xiang et al., 2025) baselines, † denotes combination with AnyDoor (Chen et al., 2024), and ‡ denotes combination with InsertAnything (Song et al., 2026). ME denotes Matching Error.

| | Method | Image Fidelity | | | Identity | | Pose |
|---|---|---|---|---|---|---|---|
| | | PSNR↑ | SSIM↑ | LPIPS↓ | CLIP↑ | DINO↑ | ME↓ |
| SD | Object3DIT† | 19.24 | 0.776 | 0.319 | 0.889 | 0.815 | 135.7 |
| | TRELLIS† | 19.51 | 0.778 | 0.312 | 0.895 | 0.848 | 75.4 |
| | Ours (SD) | **21.66** | **0.829** | **0.206** | **0.937** | **0.913** | **21.4** |
| FLUX | Object3DIT‡ | 21.32 | 0.839 | 0.225 | 0.916 | 0.857 | 98.9 |
| | TRELLIS‡ | 22.00 | 0.843 | 0.217 | 0.935 | 0.902 | 19.6 |
| | Ours (FLUX) | **23.09** | **0.871** | **0.147** | **0.959** | **0.936** | **17.8** |

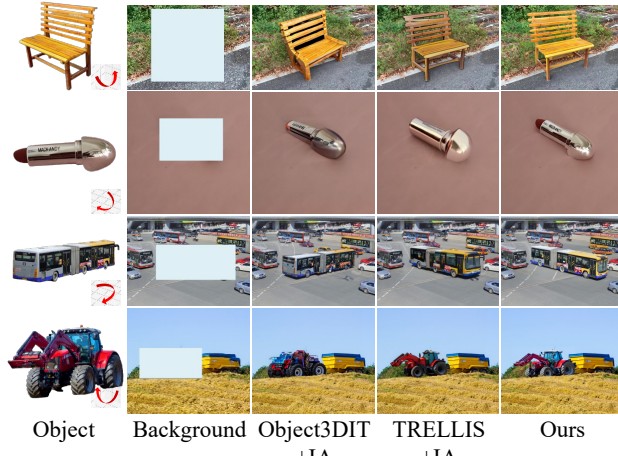

| Object | Background | Object3DIT +IA | TRELLIS +IA | Ours |
|---|---|---|---|---|

*Figure 6.* **Qualitative Comparison.** We compare our method against Object3DIT (Michel et al., 2023) and TRELLIS (Xiang et al., 2025). Our method achieves superior identity preservation and background consistency, avoiding the appearance artifacts observed in TRELLIS and the geometric distortions in Object3DIT. IA denotes InsertAnything (Song et al., 2026).

ity and perceptual similarity, we report PSNR, SSIM, and LPIPS (Zhang et al., 2018) on the full image. To evaluate reference identity preservation, we compute CLIP-I (Radford et al., 2021) and DINO (Caron et al., 2021) scores. They are calculated as the cosine similarity between feature embeddings of the ground truth and the generated image, using CLIP-ViT-B/32 and DINO-ViT-S/16 backbones, respectively. To quantify how well the generated object follows the specified pose, we further introduce a dense matching-based metric, Matching Error. Specifically, within the masked object region, we use MASt3R (Leroy et al., 2024) to establish dense correspondences between the generated object and the resized geometric condition, and compute the average pixel error over matched points. A lower Matching Error indicates more accurate adherence to the desired pose.

### 4.1. Quantitative Evaluation

Table 1 summarizes the quantitative results for different metrics with different backbones. **(1)** A consistent superiority is observed across both Stable Diffusion and FLUX-based categories on all metrics. This demonstrates the generalization ability of our unified approach regardless of the underlying generative backbone. **(2)** The improvement in reconstruction metrics (PSNR, SSIM, LPIPS) stems from our context guidance. By explicitly modeling the scene context, our framework captures environmental cues, ensuring the inserted object harmonizes with the environment. **(3)** The gains in identity metrics (CLIP-I, DINO score) highlight the effectiveness of our appearance guidance. While Object3DIT is limited by synthetic domain gaps and TRELLIS introduces texture degradation during 3D reconstruction, our approach preserves high-frequency details through real-world training and reference conditioning. **(4)** The lower Matching Error further verifies the pose accuracy of our method. The geometric condition rendered from the 3D proxy provides explicit and fine-grained pose guidance. By injecting it through a dedicated branch, our framework enables the generated object to better follow the user-specified

pose while maintaining realistic scene integration.

### 4.2. Qualitative Evaluation

Fig. 6 visually compares our method against the FLUX-based baselines. Object3DIT shows limited 3D awareness and pose control. For simple objects (Rows 1-2), it manages basic orientation changes but struggles to preserve identity. On more complex geometries (Rows 3-4), it either fails to execute the pose edit or suffers from structural collapse. TRELLIS achieves better geometric alignment through explicit 3D reconstruction but introduces significant appearance degradation. Relying exclusively on the 3D-rendered proxy frequently leads to blurry textures or deviation from the reference's unique features, making results unrealistic. In contrast, our method leverages the Geometry-Appearance-Context triplet to synthesize high-fidelity results that preserve object identity, follow precise pose transformations, and harmonize photorealistically with the background. We provide more visual demonstrations across diverse categories and scenes in Sec. H of the Appendix.

### 4.3. Pose-Change Analysis

***Effect of Pose-Change Magnitude.*** To further examine how pose variation affects generation quality, we stratify the benchmark into bins according to the VLM-annotated approximate relative rotation angles between the input and target poses. Table 2 reports the results for each bin. **(1)** The benchmark covers diverse pose changes, including a substantial portion of samples with large pose variations. **(2)** The model shows no clear degradation trend as the pose-change magnitude increases, maintaining stable image fidelity, iden-

*Table 2.* **Effect of pose-change magnitude.** We report performance under different approximate relative rotation ranges. Performance remains stable across bins. ME denotes Matching Error.

| Relative Rotation | Distribution Ratio | Image Fidelity SSIM↑ | Identity CLIP-I↑ | Pose ME↓ |
|---|---|---|---|---|
| 0–45° | 14.0% | 0.864 | 0.967 | 18.1 |
| 45–90° | 25.0% | 0.881 | 0.957 | 19.4 |
| 90–135° | 44.0% | 0.864 | 0.959 | 17.8 |
| 135–180° | 17.0% | 0.877 | 0.956 | 15.7 |
| Overall | 100.0% | 0.871 | 0.959 | 17.8 |

tity preservation, and pose accuracy across different pose-change ranges. This suggests that our method remains effective under moderate and large pose variations.

***Performance under Large Pose Changes.*** We further visualize representative large pose-change examples in Fig. 7. These examples cover challenging cases such as substantial rotations, unseen-view synthesis from limited reference appearance, and counterfactual pose changes. The results show that our method can handle substantial pose changes while maintaining pose consistency and appearance fidelity.

### 4.4. Further Analysis

***Effectiveness of RGB Geometry Guidance.*** As discussed in Sec. 3.2, standard geometric signals such as normal maps often suffer from semantic ambiguity for symmetric objects. We empirically verify this by comparing our RGB-based guidance against a surface normal baseline. The results, illustrated in Fig. 8, confirm that while normal maps effectively capture the physical silhouette, they fail to distinguish the semantic orientation of a circular road sign. In contrast, our guidance retains semantically rich textural cues, which enable the generator to resolve rotational ambiguity and enforce precise semantic alignment.

***Importance of Decomposed Injection.*** We investigate the mechanism for integrating the visual triplet. Naive concatenation of multiple guidance signals often leads to feature entanglement, as shown in Fig. 9. Without decomposed pathways, the model tends to over-rely on the appearance of the geometry guidance, resulting in blurry textures. In contrast, our decomposed strategy successfully isolates structural guidance from identity preservation.

***Quantitative Component Analysis.*** For the ablation study, the baseline uses decomposed injection with appearance and geometry guidance only. Results in Table 3 show monotonic improvements across all metrics, validating the cumulative effectiveness of our framework. **(1)** Training on hybrid data brings the most significant improvements in identity preservation and pose accuracy, raising CLIP-I from 0.904 to 0.943 and reducing Matching Error from 26.9 to 22.7. This indicates that relying solely on existing multi-

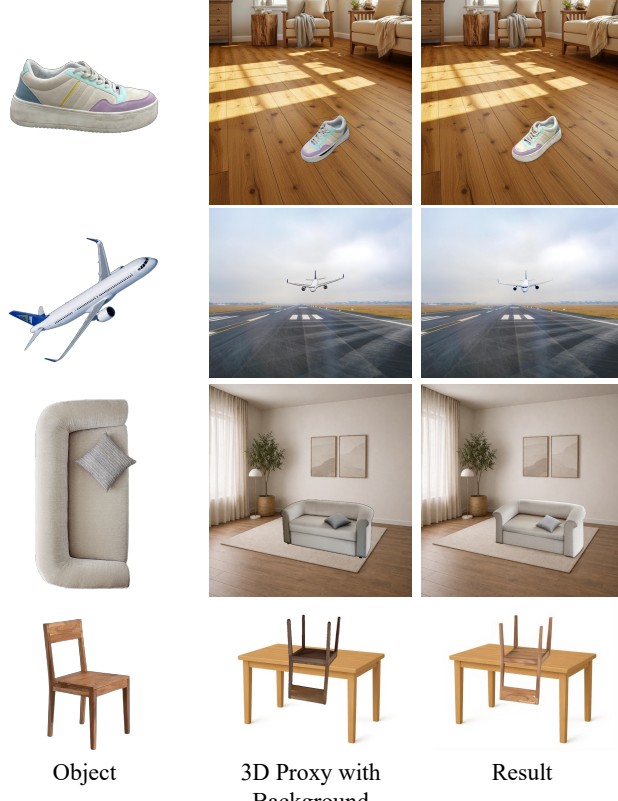

Object     3D Proxy with Background     Result

*Figure 7.* **Large pose-change examples.** Representative cases show substantial pose variations between the reference object and target pose. These examples require synthesis of largely unseen object views from limited reference appearance, including large rotations, top-view to side-view transformation, and near 180° viewpoint changes. Our method preserves object identity while following the specified pose.

view datasets limits generalization to diverse objects and poses in real-world scenes. **(2)** Integration of context guidance further improves reconstruction fidelity (PSNR ↑ 0.18), suggesting the importance of global scene awareness for foreground-background harmonization. **(3)** Notably, Shape-Decomposed Mask Augmentation is critical for perceptual quality, reducing LPIPS from 0.190 to 0.155 and Matching Error from 20.7 to 19.0. We attribute this to reduced reliance on boundary artifacts, encouraging the model to better leverage the geometric condition and learn more robust internal texture representations. **(4)** Finally, progressive resolution training ensures generalization to high-resolution images, culminating in the best performance across all metrics.

***Robustness Against 3D Reconstruction Degradation.*** A key advantage of our framework is its robustness to quality degradation in the intermediate 3D proxy. As shown in Fig. 10, generative 3D reconstruction models often struggle to preserve high-frequency semantic details, causing artifacts such as blurred or distorted surface text. Nevertheless, our model generates clear and legible text in the final output,

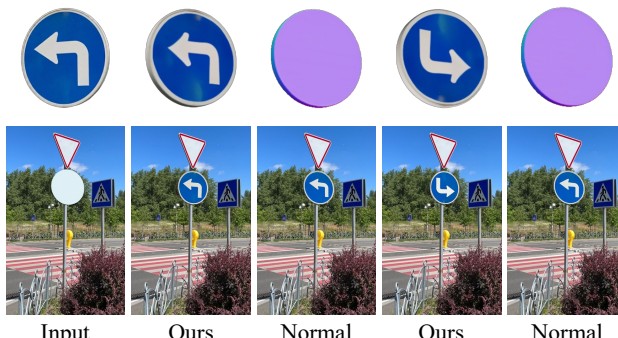

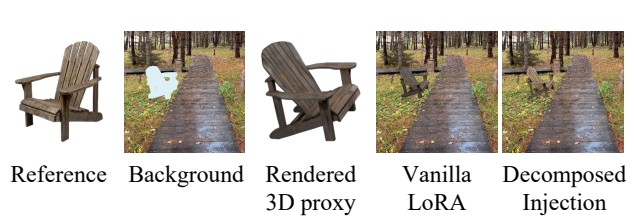

| Input | Ours | Normal | Ours | Normal |

*Figure 8.* **Comparison of geometry guidance signals.** Top row: Reference object, RGB/normal guidance at $0°$, and RGB/normal guidance at $180°$. Bottom row: Background image and the four corresponding generation results. For the symmetric road sign, the normal maps are invariant to the $180°$ rotation, leading to semantic ambiguity and orientation errors in the normal-based results. In contrast, our RGB proxy provides semantically rich textural cues, ensuring the model follows the desired pose.

| Reference | Background | Rendered 3D proxy | Vanilla LoRA | Decomposed Injection |

*Figure 9.* **Effectiveness of the decomposed injection.** We compare our approach against a vanilla LoRA baseline that naively concatenates the appearance and geometry guidance. When the 3D proxy contains texture artifacts, the vanilla baseline suffers from feature entanglement, incorrectly inheriting degraded details. Our decomposed strategy successfully isolates these conflicting signals, leveraging the proxy for geometry guidance while preserving high-fidelity identity from the reference.

correcting errors in the 3D condition. This capability validates our decomposed injection strategy: by utilizing the 3D proxy mainly for geometry guidance while retrieving texture information directly from the high-quality reference image, our method preserves complex visual semantics even when 3D reconstruction is degraded.

***Failure Case Analysis.*** Since our method strictly adheres to the geometry guidance from the 3D proxy, its performance is inevitably bounded by the upstream image-to-3D reconstruction. Although our method can mitigate 3D reconstruction degradation in general cases, when 3D generation fails to capture the correct coarse geometry, such as severe aspect ratio distortion, these errors may propagate into the final output. As shown in Fig. 11, for a rectangular plaque, the image-to-3D model incorrectly reconstructs it as a square. Our generator synthesizes the object within this distorted square boundary and fails to recover the original rectangular shape. This illustrates a trade-off: while strict geometric adherence enables precise pose control, it also requires a reasonably accurate 3D proxy as the starting point.

*Table 3.* **Ablation study.** The baseline only uses the decomposed injection strategy with appearance and geometry guidance. Hybrid Data: training on the curated real-world dataset. Context: integration of context guidance. Mask Aug.: Shape-Decomposed Mask Augmentation strategy. Progressive: progressive resolution training. ME denotes Matching Error.

| Configuration | Image Fidelity | | | Identity | | Pose |
|---|---|---|---|---|---|---|
| | PSNR↑ | SSIM↑ | LPIPS↓ | CLIP↑ | DINO↑ | ME↓ |
| Base | 22.26 | 0.866 | 0.207 | 0.904 | 0.915 | 26.9 |
| + Hybrid Data | 22.56 | 0.868 | 0.192 | 0.943 | 0.930 | 22.7 |
| + Context | 22.74 | 0.869 | 0.190 | 0.952 | 0.932 | 20.7 |
| + Mask Aug. | 22.89 | 0.869 | 0.155 | 0.957 | 0.935 | 19.0 |
| + Progressive | **23.09** | **0.871** | **0.147** | **0.959** | **0.936** | **17.8** |

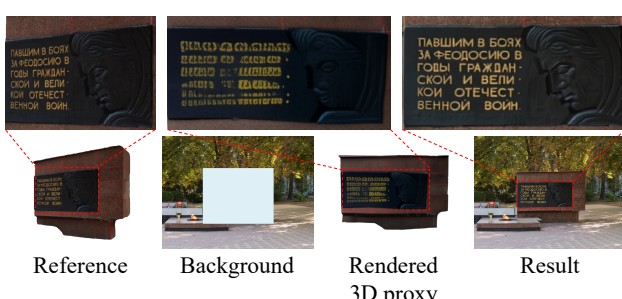

| Reference | Background | Rendered 3D proxy | Result |

*Figure 10.* **Robustness to degraded 3D proxies.** In an extreme object insertion case with rich textual details on the object surface, the 3D proxy suffers from significant quality degradation. In contrast, our model inserts precise, legible details.

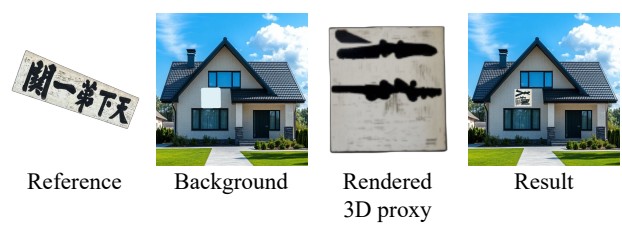

| Reference | Background | Rendered 3D proxy | Result |

*Figure 11.* **Failure case.** The upstream model incorrectly reconstructs the rectangular reference as a square proxy. Our model strictly follows this distorted geometric condition, resulting in an incorrect aspect ratio in the final output.

Appendices E–G provide additional analyses on latency, proxy-scene misalignment, and complex environments.

# 5. Conclusions

In this work, we present DIRECT, a framework for pose-controllable object insertion. By decomposing conditioning signals into a visual triplet (geometry, appearance, and context) and injecting them through independent pathways, DIRECT reconciles 3D spatial control with high-fidelity 2D synthesis, achieving state-of-the-art performance. Future work will explore end-to-end geometry refinement during generation to reduce severe proxy topology errors, further advancing 3D-aware image editing.

## Acknowledgements

This research was partially supported by NSFC (NO. 62225604), Shenzhen Science, Technology Program (JCYJ20240813114237048) and the Zhongguancun Academy (Grant No. C20250604). This research was also supported by cash and in-kind funding from NTU S-Lab and industry partner(s).

## Impact Statement

This paper presents a framework for high-fidelity, controllable object insertion, contributing to the fields of generative media and augmented reality. Our method lowers the technical barrier to complex image composition, offering significant benefits to applications such as virtual staging, e-commerce photography, and creative design. However, as with any technology capable of generating photorealistic manipulations, there is a potential risk of misuse in creating misleading visual content. The ability to seamlessly insert objects with precise 3D geometric control could theoretically be exploited to alter visual evidence or fabricate scenarios. We advocate for the responsible development and deployment of such models, including the integration of digital watermarking and provenance tracking technologies.

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

# Appendix Overview

This appendix provides prompts for competing methods, data construction details, implementation details, additional baselines, extended analyses, and visual demonstrations.

Sec. A: Prompts used for competing methods in Fig. 1.
Sec. B: Training data construction details.
Sec. C: Interactive inference pipeline and derivation of explicit geometric conditions.
Sec. D: Additional comparison with an intrinsic-guided compositing baseline.
Sec. E: Inference latency and memory overhead analysis.
Sec. F: Sensitivity analysis under 3D proxy-scene misalignment.
Sec. G: Performance in complex environments involving occlusion, lighting, and reflections.
Sec. H: Additional visual demonstrations across diverse objects and real-world scenes.

# A. Prompts for Competing Methods in Fig. 1

**Nano Banana Pro (Google, 2025).** Insert the book from the first image into the bookshelf in the second image. Place the book on the second shelf from the top, leaning against the books on the right side.

**Object3DIT (Michel et al., 2023).** Rotate the book by $320°$.

# B. Training Data Construction Details

***Strategies to ensure diversity and quality.*** Our training data combines automatically curated pairs from SA-1B (Kirillov et al., 2023) with filtered multi-view data from MVImgNet (Yu et al., 2023). We discuss the strategies used to ensure diversity and quality for these two sources separately.

- **Generated pairs from SA-1B.** We impose explicit quality-control constraints at each stage of the automatic construction pipeline.
  1. **Object image curation.** The curation stage aims to retain images containing fully visible, unoccluded objects with precise object masks.
     - In the propose step, the VLM (Bai et al., 2025) agent identifies salient object categories in the scene and excludes images without suitable foreground objects.
     - In the segment step, SAM-3 (Carion et al., 2025) produces candidate masks for the proposed instances. We filter out objects whose masks touch the image boundary or are too small, thereby improving object completeness and visual clarity.
     - In the verify step, the agent performs a zoom-in check on a local crop to further assess both structural completeness and boundary precision, discarding truncated, occluded, or poorly segmented instances.
  2. **Reference image synthesis.** We use the verified mask from the curation stage to remove the background and employ an image editing model (Wu et al., 2025) to synthesize a novel-view reference image. This allows the editing model to focus on generating the object itself. Meanwhile, the original real image is always used as the supervision target, keeping the training signal anchored to real-world captures with realistic object-scene interactions.
- **MVImgNet.** We use Tenengrad (Tenenbaum, 1971), CLIP-IQA (Wang et al., 2023), and Q-Align (Wu et al., 2023) scores to filter out blurry and low-quality samples before incorporating them into the hybrid dataset.

***Verified benefit.*** The benefit of the hybrid training data is empirically verified in Table 3. Training with hybrid data significantly improves both identity preservation and pose accuracy.

***Remaining limitations.*** Despite these efforts, since our dataset is constructed or filtered from existing source datasets, it may still inherit some of their biases, such as category imbalance and dataset-specific collection bias. We aim to minimize avoidable construction errors through explicit filtering and verification. However, residual errors may still remain due to imperfect masks, synthesis artifacts, or occasional failures in the automatic curation process.

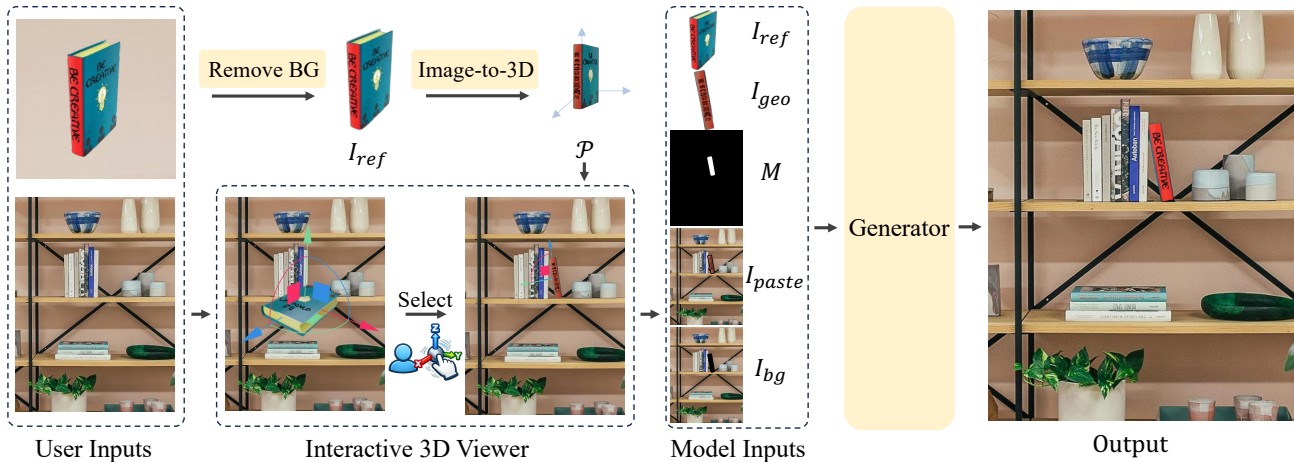

*Figure 12.* **Overview of the Interactive Inference Pipeline.** First, the reference image is lifted into a 3D proxy. Users then manipulate the proxy over the background canvas via a visual gizmo to determine the target 6-DoF pose. Finally, the system automatically renders the necessary conditions to guide our generative framework, yielding a high-fidelity composite image that respects the user-specified pose.

## C. Interactive Inference Pipeline

We provide a **video demo** on the project page showing the full inference process.

To instantiate the formulation defined in Sec. 3.1, we provide an interactive alignment interface (Fig. 12) that converts user intent into explicit geometric constraints. Instead of requiring transformation matrices or manual mask annotation, users specify the target pose by manipulating a coarse 3D proxy $\mathcal{P}$ over the background. The finalized alignment deterministically yields the 6-DoF pose $\xi$ and the insertion region $M$, producing pixel-aligned conditions for our generator. The workflow consists of two stages: proxy lifting and condition derivation.

**Lifting the reference object into an interactive 3D proxy.** Given the reference object image, we employ a foreground segmentation model (Zheng et al., 2024) to remove the background, obtaining the clean reference object image $I_{ref}$. Then $I_{ref}$ is processed by the image-to-3D model TRELLIS (Xiang et al., 2025) to generate a 3D visual proxy $\mathcal{P}$, represented as a set of 3D Gaussians. To enable precise manipulation without requiring specialized 3D knowledge, we integrate $\mathcal{P}$ into an interactive 3D viewer built on Viser (Yi et al., 2025). The background image is set as a static canvas, and a visual transform-control gizmo is attached to $\mathcal{P}$. This interface allows users to effortlessly translate and rotate the object, aligning its 6-DoF pose $\xi$ with the desired region and perspective within the scene.

**Deriving conditions from the interaction.** Once the interaction is finalized, the system automatically derives the inputs for the generative model. First, the proxy $\mathcal{P}$ is rendered at pose $\xi$ to obtain the rendered image $I_{render}$ and its binary alpha mask $m$. A composite background $I_{paste}$ is then created by overlaying $I_{render}$ onto the background within a dilated mask region, with the corresponding inpainting mask denoted as $M$. Separately, the object is recentered within $I_{render}$ to obtain $I_{geo}$, which provides explicit geometry information. Finally, the set $(I_{ref}, I_{geo}, I_{paste}, I_{bg}, M)$ serves as the input to our generative model, where $I_{paste}$ is subsequently cropped to form the local composite $I_{local}$ described in Sec. 3.2.

## D. Additional Intrinsic-Guided Compositing Baseline

We additionally evaluate an alternative class of methods for pose-controllable object insertion based on intrinsic-guided compositing. These methods typically take a 3D asset as input and perform image compositing using intrinsic maps from both the asset and the target scene as conditions. To adapt this paradigm to our setting, we first reconstruct a 3D asset from the reference object using TRELLIS (Xiang et al., 2025), and then use an intrinsic-guided compositing method to insert the asset into the target scene. Specifically, we use ZeroComp (Zhang et al., 2025) as a representative method of this class and evaluate the resulting TRELLIS+ZeroComp baseline both quantitatively and qualitatively.

As shown in Table 4, this intrinsic-guided compositing baseline achieves a very low Matching Error, reflecting strong

adherence to the rendered proxy. This is expected, as the compositing process directly leverages intrinsic maps as conditions. However, it performs substantially worse in image fidelity and identity preservation, as these conditions do not explicitly preserve the reference object's appearance. By contrast, our goal is to insert the reference object with both pose control and strong appearance preservation. Under this task requirement, our method performs better overall. This trade-off is also evident in the qualitative comparisons in Fig. 13. The baseline follows the rendered asset closely, but often loses fine-grained reference appearance, whereas our method better preserves object identity while producing more realistic insertion results.

*Table 4.* **Additional intrinsic-guided compositing baseline.** TRELLIS+ZeroComp achieves low Matching Error via direct intrinsic guidance from the asset and target scene, but performs worse in image fidelity and identity preservation. ME denotes Matching Error.

| Method | Image Fidelity | | | Identity | | Pose Accuracy |
|---|---|---|---|---|---|---|
| | PSNR↑ | SSIM↑ | LPIPS↓ | CLIP-I↑ | DINO↑ | ME↓ |
| TRELLIS + ZeroComp | 20.66 | 0.816 | 0.297 | 0.892 | 0.828 | **5.2** |
| Ours (SD) | 21.66 | 0.829 | 0.206 | 0.937 | 0.913 | 21.4 |
| Ours (FLUX) | **23.09** | **0.871** | **0.147** | **0.959** | **0.936** | 17.8 |

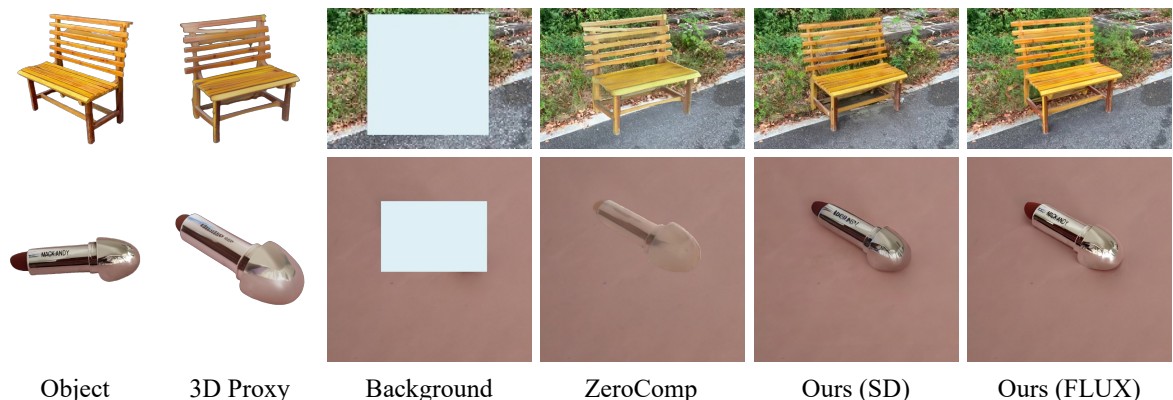

| Object | 3D Proxy | Background | ZeroComp | Ours (SD) | Ours (FLUX) |

*Figure 13.* **Qualitative comparison with intrinsic-guided compositing.** The intrinsic-guided compositing baseline provides strong geometric adherence, but struggles to preserve fine-grained reference appearance and overall image realism. In contrast, our method simultaneously achieves pose control, identity preservation, and realistic scene integration.

## E. Inference Latency and Memory Overhead

Since our framework introduces an additional 3D proxy generation component, it is important to analyze the resulting computational overhead. We therefore evaluate the inference latency and memory overhead in the SD-based setting, comparing our method with the corresponding SD-based baselines. We break down the runtime into the 3D generation stage, the 2D generation stage, and other processing steps, and also report the overall end-to-end latency and peak allocated GPU memory.

As shown in Table 5, our method achieves end-to-end latency and peak memory usage comparable to those of the baselines. Our 3D generation stage is slower than Object3DIT, but this cost is incurred upfront before interaction, since the user starts specifying the pose and insertion region after the 3D proxy has been generated. By contrast, our 2D generation stage is faster than the other methods compared. Overall, this leads to competitive end-to-end latency.

*Table 5.* **Inference latency and memory overhead.** We report the runtime breakdown, overall end-to-end latency, and peak allocated GPU memory in the SD-based setting.

| Method | 3D (s) | 2D (s) | Other (s) | Overall (s) | Mem. (GB) |
|---|---|---|---|---|---|
| Object3DIT | 2.143 | 6.031 | 0.436 | 8.610 | 9.987 |
| TRELLIS | 5.054 | 6.031 | 0.412 | 11.500 | 11.790 |
| Ours | 5.054 | 4.212 | 0.276 | 9.542 | 10.047 |

## F. Sensitivity to 3D Proxy-Scene Misalignment

The accuracy of the user-specified 3D proxy placement relative to the target scene is an important factor for practical object insertion. In real user interactions, the proxy may not be perfectly aligned with the surrounding scene geometry due to imperfect manipulation or ambiguity in the desired placement. To examine the sensitivity of our method to such proxy-scene misalignment, we provide representative examples with mild placement inaccuracies in Fig. 14. The results show that our method can compensate for small misalignments between the proxy and the scene, producing natural insertion results. This suggests that our framework does not require users to specify a perfectly precise proxy-scene alignment, making the interaction more tolerant and user-friendly.

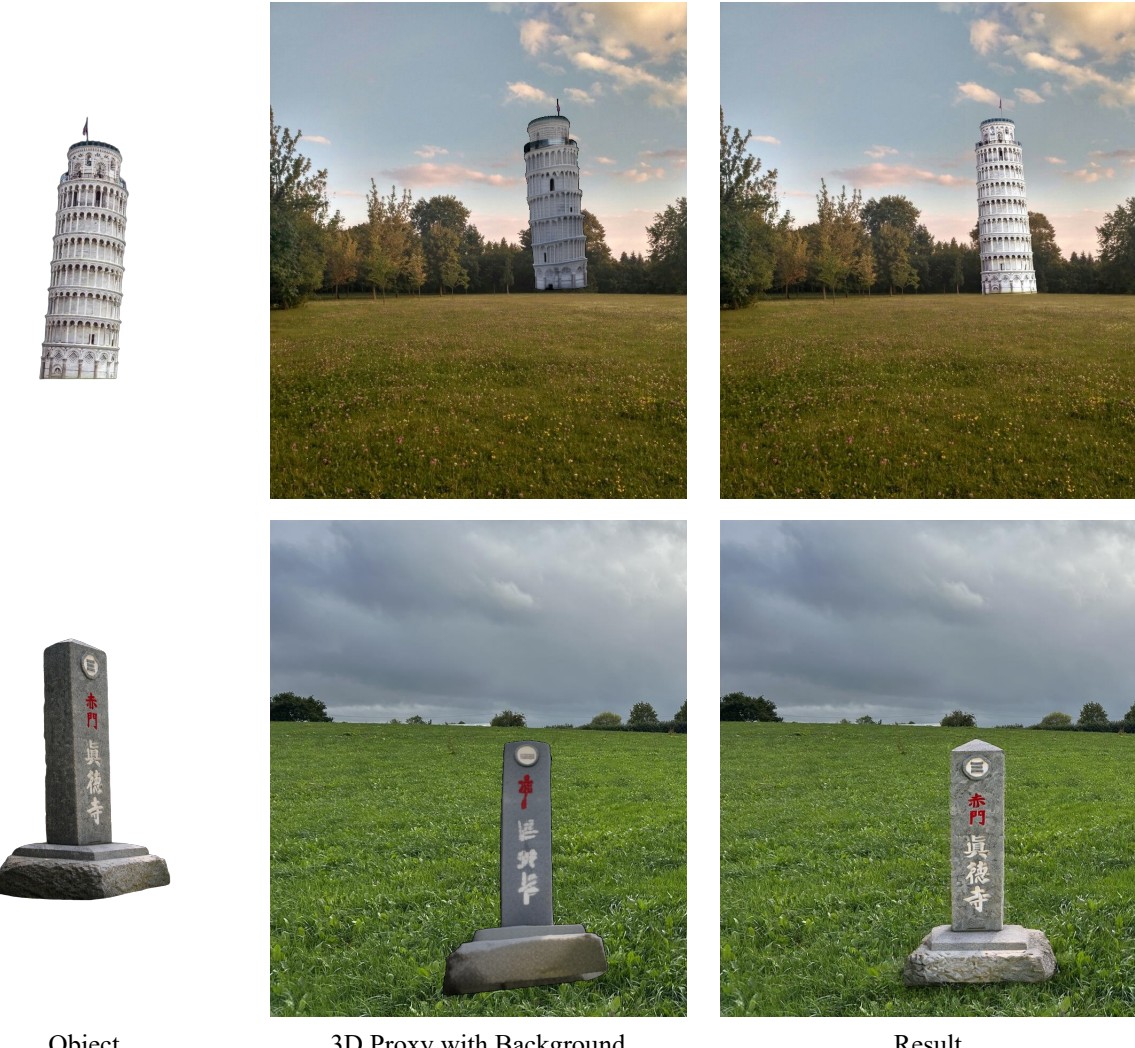

Object        3D Proxy with Background        Result

*Figure 14.* **Sensitivity to 3D proxy-scene misalignment.** We show representative cases where the user-specified 3D proxy is mildly misaligned with the target scene. In the first example, the proxy is placed slightly above the ground. In the second example, the proxy is not perfectly aligned with the supporting surface. Despite these mild proxy-scene placement errors, our method produces natural insertion results, suggesting robustness to small inaccuracies in user-specified proxy placement.

# G. Performance in Complex Environments

Real-world object insertion often involves complex scene effects, such as occlusion, directional lighting, and reflections. Although our method does not explicitly model physical interactions, illumination, or view-dependent material effects, it can still produce visually plausible results in these challenging scenarios. We provide representative examples in Fig. 15. These results demonstrate that the learned context guidance helps the model infer plausible object-scene interactions, even without explicit physical simulation.

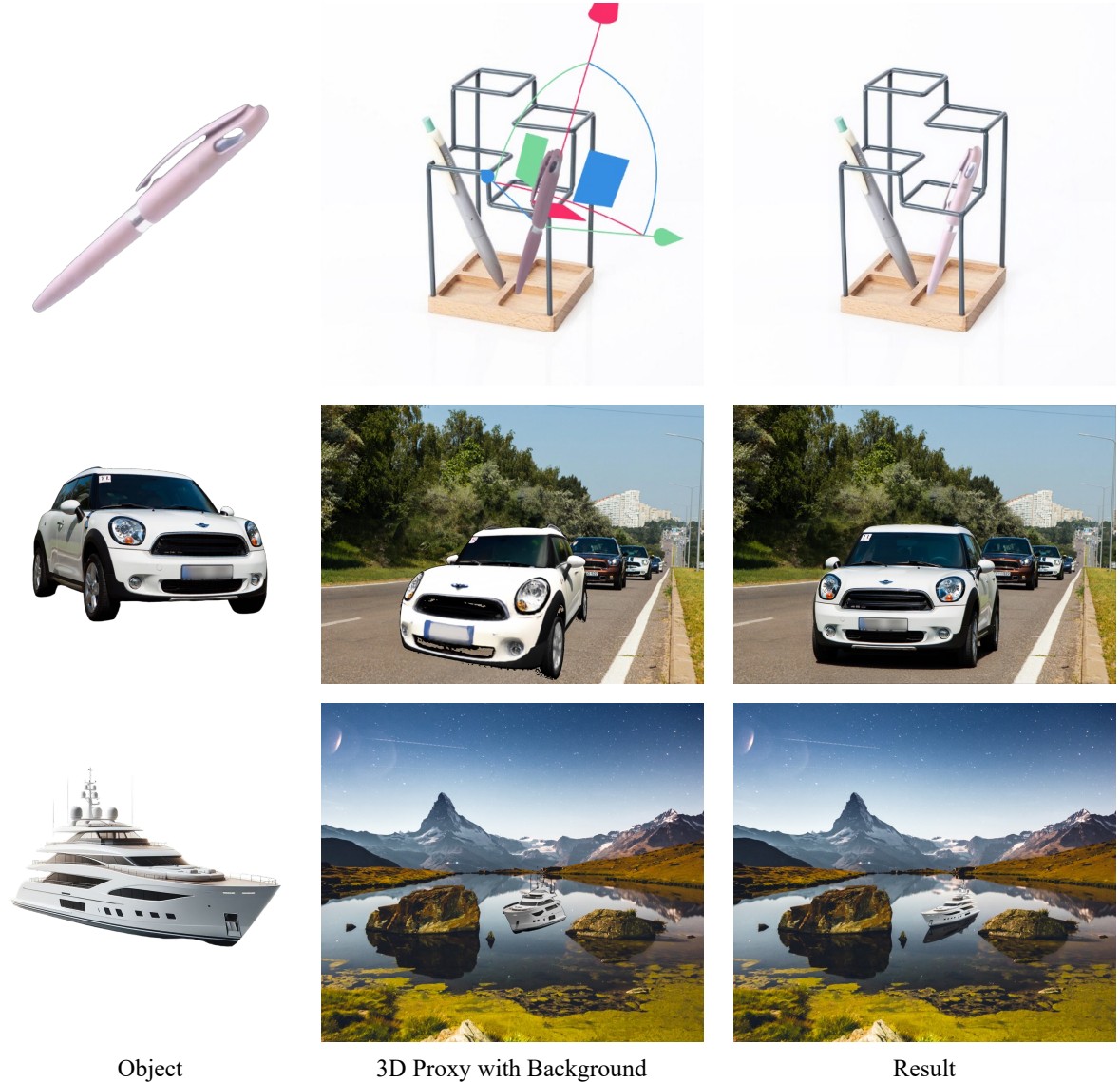

| Object | 3D Proxy with Background | Result |

*Figure 15.* **Performance in complex environments.** We show representative examples involving occlusion, lighting, and reflection. For occlusion, a pen is inserted into a pen holder, where the generated result exhibits a plausible depth relationship between the pen and the holder structure. For lighting, a car is inserted into a scene with strong directional illumination, and the model generates a plausible shadow consistent with the surrounding scene. For reflection, a boat is inserted onto a reflective water surface, and the generated result includes a visually plausible reflection on the background surface.

# H. Visual Demonstrations

In this section, we provide more visual examples in Fig. 16 to further demonstrate the robustness and generalization of DIRECT.

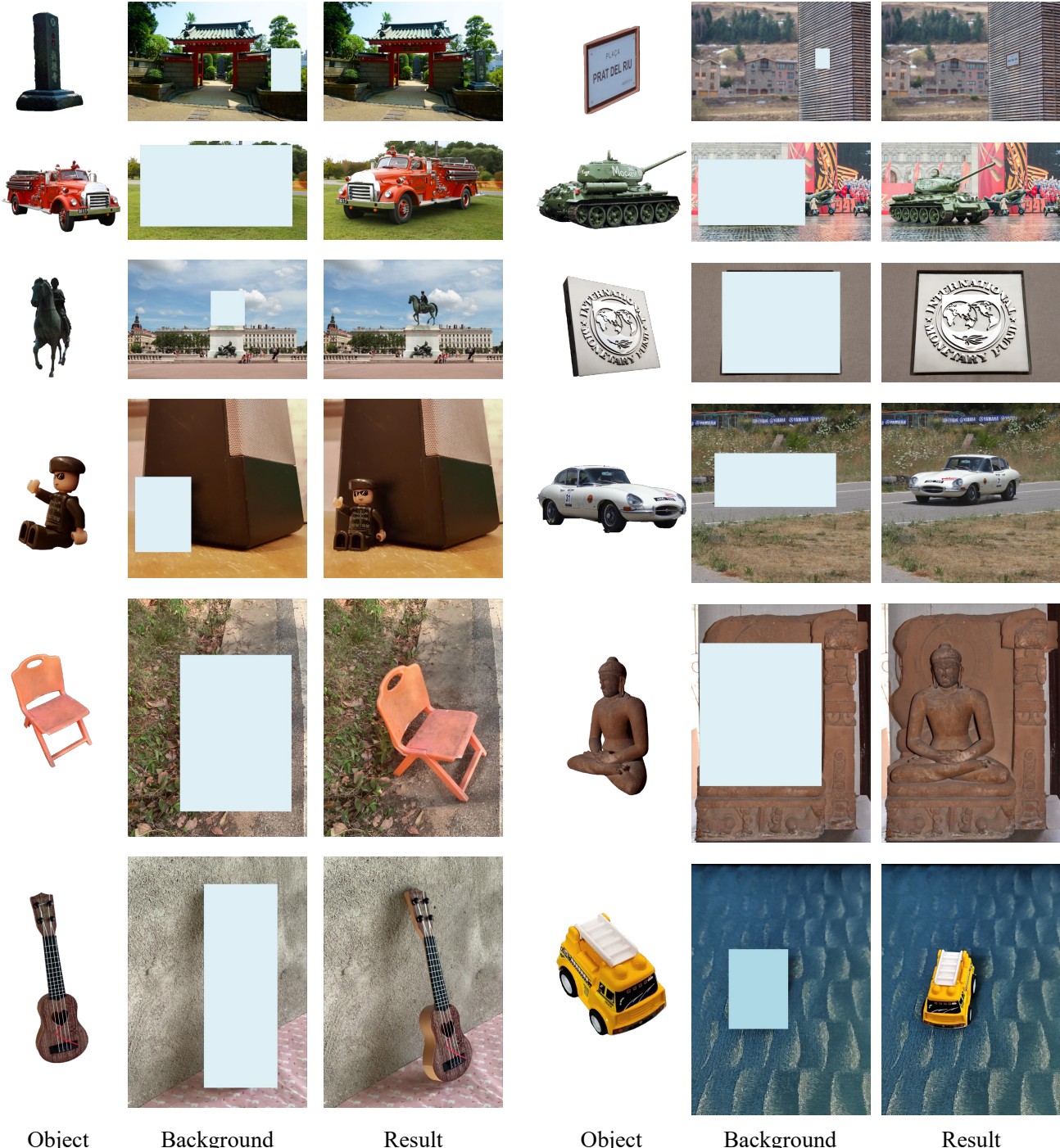

| Object | Background | Result | Object | Background | Result |

*Figure 16.* **Visual Demonstrations.** We showcase our model's capability to insert various objects into complex real-world backgrounds with high visual fidelity. The results show that our method supports explicit pose control (e.g., varying angles and orientations) while strictly preserving the identity and texture details of the reference objects.

