# OpenReview forum: "Direct 3D-Aware Object Insertion via Decomposed Visual Proxies"
_ICML.cc/2026/Conference — ICML 2026 regular_

### Official Review · Reviewer_e7G4 · 2026-02-24

**Soundness:** 3
**Presentation:** 2
**Significance:** 2
**Originality:** 2
**Overall Recommendation:** 4
**Confidence:** 3

**Summary:**

This work proposes a method for 3D-aware object insertion into images. Specifically, this work enables pose-controlled inputs into images, where a 3D proxy is controlled by the user. The inputs to the model are appearance, geometry, and context guidance, which are available in the dataset curated by the authors. Experiments are performed on the curated test set.

**Compliance With Llm Reviewing Policy:**

Affirmed.

**Final Justification:**

As my concerns have been satisfactorily addressed in the author's response, and I have increased my score to "Weak Accept".

**Key Questions For Authors:**

Please refer to weaknesses. I currently recommend a “Weak Reject”, but will consider raising the score if my concerns are adequately addressed.

**Limitations:**

Failure cases have been discussed.

**Strengths And Weaknesses:**

**Strengths**

In my opinion, this paper tackles a meaningful problem of inserting objects into images with controlled poses. I recognize that previous methods are often limited in that aspect.

**Weaknesses**

I find the training strategy part not well explained, particularly the geometric alignment part. It is hard to visualize all the steps, including the  rendering of predefined poses, and then the 3DGS-bsaed refinement. These should be clarified, ideally with a schematic.

Also, at inference time, the users should be able to interact with the visual gizmo (e.g., in Fig 10 in supplementary). I would like more information about this visual gizmo, e.g. does it provide a drag-and-rotate interface for the reconstructed 3D object?

The reported metrics also currently seem somewhat incomplete. The reported metrics such as PSNR, SSIM and LPIPS measure image fidelity, and CLIP /DINO metrics measure identity. However, there are no metrics that actually measure the accuracy of the generated images in terms of the input poses. Thus, the claim and quality of “pose-controllable” object insertion as not actually been evaluated.

The above point is rather important to me, since the paper claims to have a pose-controllable object insertion, yet these have not been truly evaluated at all. I suggest the authors to add additional metrics, and to use some synthetic datasets to evaluate this if needed.

I also feel that the evaluation is currently a bit lacking, especially in terms of poses that are very different from the input pose. For instance, it could be that most of the poses in the dataset do not require large changes in the poses of the objects. Specifically, I would like to see the evaluation of input poses that are very different from the input poses. How much is the performance affected by the input poses?

The number of baseline methods reported is 2. Essentially, only object3dit is truly a 3d-aware object insertion model. I wonder if the authors can run more recent baselines on their evaluation set as well.

---

> ### Author Rebuttal · Authors · 2026-03-31
>
> Anonymous figures for this response are provided here: https://anonymous.4open.science/api/repo/ICML21834_Rebuttal_e7G4/file/figures.pdf?v=34e8d5c1
>
> **W1. Schematic of Pose Alignment.**
>
> We thank the reviewer for this helpful suggestion. To make this part clearer, we have added a schematic with intermediate visualizations for each step of the geometric alignment pipeline, shown as Fig. a in the link. We hope this figure, together with the description in the paper, clarifies the overall procedure. We would be glad to further clarify any remaining ambiguity if helpful.
>
> ---
>
> **W2. Interactive Inference.**
>
> Yes. A user-friendly interaction interface is an essential component of Direct 3D-Aware Object Insertion. During inference, the user can directly manipulate the 3D proxy through a drag-and-rotate interface. Specifically, the user is presented with the interface shown in Fig. 10 and can easily use the mouse to drag, rotate, and adjust the object to obtain the desired placement region and pose. After the pose is specified through this interaction, the corresponding geometric condition is rendered and used for insertion.
>
> We also provide a video demo in the supplementary material, which shows the actual inference process with user interaction workflow.
>
> ---
>
> **W3. Pose Controllability Evaluation.**
>
> We thank the reviewer for raising this important point. We agree that evaluating pose accuracy is an important aspect of validating the effectiveness of our method.
>
> We adopted a dense matching-based evaluation. Specifically, we remove the background from $I_{out}$, resize it to the same resolution as $I_{geo}$, and use MASt3R to estimate dense correspondences on the object regions between the two images. We then compute the average pixel error over the matched points.
>
> |Group|Method|Match Error ↓|
> |---|---|---:|
> |FLUX|Object3DIT + IA|98.9|
> |FLUX|TRELLIS + IA|19.6|
> |FLUX|Ours (FLUX)|**17.8**|
> |SD|Object3DIT + AD|135.7|
> |SD|TRELLIS + AD|75.4|
> |SD|Ours (SD)|**21.4**|
>
> Our method achieves the best overall performance in both the SD-based and FLUX-based categories.
>
> ---
>
> **W4. Performance on Large Pose Change.**
>
> We thank the reviewer for raising this important point.
> To better understand how our method behaves under different pose changes, we stratify the benchmark into bins of VLM-annotated approximate relative rotation angles between the input and target poses. We then report the results for each bin below.
>
> |Rotation Angle Range|0-45|45-90|90-135|135-180|Overall|
> |---|---:|---:|---:|---:|---:|
> |Sample Ratio|14.0%|25.0%|44.0%|17.0%|100.0%|
> |Image Fidelity (SSIM) ↑|0.864|0.881|0.864|0.877|0.871|
> |Identity (CLIP) ↑|0.967|0.957|0.959|0.956|0.959|
> |Pose Accuracy (Match Error) ↓|18.1|19.4|17.8|15.7|17.8|
>
> (1) The benchmark covers a relatively diverse range of pose-change bins. (2) We do not observe a clear degradation trend as the pose change increases. Image fidelity, identity consistency, and pose accuracy remain stable across all bins.
>
> To further evaluate performance under large pose changes, we provide four representative examples in Fig. b in the link:
>
> - The first two examples show typical large pose changes.
> - The third example starts from a top-view input object and generates a side-view result.
> - The fourth example applies a 180° rotation and requires a counterfactual generation result.
>
> In all four cases, our method produces good results, suggesting that it can handle large pose changes while maintaining pose consistency and appearance fidelity.
>
> ---
>
> **W5. Additional Baselines.**
>
> We thank the reviewer for this helpful suggestion. We additionally include the recent 3D-aware insertion model ZeroComp (WACV 2025). To make it applicable to our setting, we combine TRELLIS with ZeroComp as a complete baseline, where TRELLIS provides the reconstructed 3D proxy and ZeroComp performs the insertion.
>
> |Method|PSNR ↑|SSIM ↑|LPIPS ↓|CLIP ↑|DINO ↑|Match Error ↓|
> |---|---:|---:|---:|---:|---:|---:|
> |ZeroComp|20.66|0.816|0.297|0.892|0.828|5.2|
> |Ours (SD)|21.66|0.829|0.206|0.937|0.913|21.4|
> |Ours (FLUX)|**23.09**|**0.871**|**0.147**|**0.959**|**0.936**|17.8|
>
> As shown in the table, this baseline achieves very low match error, reflecting strong adherence to the rendered proxy. This is expected, since ZeroComp uses intrinsic maps from the 3D proxy as guidance. However, it performs substantially worse on image fidelity and identity preservation, as these conditions do not explicitly preserve the reference object's appearance.
>
> By contrast, our goal is to insert the reference object with both pose control and strong appearance preservation. Under this task requirement, our method performs better overall than ZeroComp.
>
> This difference is also more directly illustrated in the qualitative comparisons shown in Fig. c in the link.

---

> > ### Author Rebuttal · Reviewer_e7G4 · 2026-04-01
> >
> > I thank the authors for their informative rebuttal. My concerns have been satisfactorily addressed, and I have increased my score as well.

---

> > > ### Author Response · Authors · 2026-04-02
> > >
> > > Thank you very much for your follow-up and for taking the time to read our rebuttal carefully. We are very glad that our response helped address your concerns.
> > >
> > > We noticed that you mentioned increasing your score in your comment, but on our side the visible score does not seem to have changed yet. This may simply be a display delay, but we just wanted to mention it in case an update was intended.
> > >
> > > Thank you again for your thoughtful feedback and consideration.

---

### Official Review · Reviewer_CUXV · 2026-02-27

**Soundness:** 3
**Presentation:** 3
**Significance:** 3
**Originality:** 3
**Overall Recommendation:** 4
**Confidence:** 3

**Summary:**

This paper proposes DIRECT, a framework for pose-controllable object insertion that decomposes conditioning signals into appearance, geometry, and context to combine explicit 3D spatial control with 2D image synthesis.

**Compliance With Llm Reviewing Policy:**

Affirmed.

**Final Justification:**

My concerns are fully addressed, so I have raised my score.

**Key Questions For Authors:**

1. It would be good to add an audio sound explanation in the supplementary video demo to better explain the interactive process.

**Limitations:**

yes

**Strengths And Weaknesses:**

Strengths:
The motivation is clear, the proposed method outperforms SoTA baselines, and the writing is easy to follow.
Weaknesses:
1. The authors claim the proposed method preserves high-frequency details and pristine object identity, yet the input reference images I_ref seen by the model during the training phase are synthesized by an editing model that is limited to generating "coarse novel views". This creates a logical contradiction regarding how the model learns high-fidelity reconstruction from degraded inputs, which fundamentally questions the soundness of the overall experimental setup.
2. The core selling point of this paper is explicit pose controllability. However, the evaluation section completely lacks metrics specifically measuring pose accuracy or geometric alignment. Relying solely on general image quality and identity metrics like PSNR, SSIM, and CLIP makes it impossible to verify the actual effectiveness of the pose control.
3. The entire inference pipeline is long and cumbersome, involving segmentation, 3D reconstruction, and multi-stage rendering. The paper lacks any evaluation of computational efficiency (e.g., inference latency, memory overhead) and fails to compare these practical costs against the baselines.
4. The authors should add ZeroComp as a baseline. Because DIRECT itself uses TRELLIS to lift the reference into a 3D asset, the authors could have easily fed the exact same TRELLIS-generated 3D proxy into ZeroComp for a fair comparison both quantitatively and qualitatively.

---

> ### Author Rebuttal · Authors · 2026-03-31
>
> **W1. Clarification of Synthesized Training Data.**
>
> Thanks for giving us the opportunity to clarify this point.
>
> **(1) The "coarse novel view" has high-fidelity objects.** The image editing model we use, Qwen-Image-Edit, has a strong consistency-preserving ability and can retain high-frequency details and pristine object identity well.
>
> **(2) The "coarse" means the pose control is not precise.** For example, given a prompt such as “Rotate the camera 90 degrees to the right,” the model may produce an image with strong identity consistency but with an actual viewpoint change closer to 60 degrees. In the paper (L230–L235), we use the term “coarse novel view” specifically to describe this inability to realize accurate pose transformations.
>
> **(3) Qwen-Image-Edit cannot solve our task, but could provide training data.** Consider these points, Qwen-Image-Edit is not suitable for directly solving our task, which requires precise pose-controllable insertion. However, it is appropriate for our data generation pipeline, because the accurate pose supervision in our training data is provided by Geometric Alignment rather than by the editing model itself.
>
> ---
>
> **W2. Pose Controllability Evaluation.**
>
> We thank the reviewer for raising this point. We agree that evaluating pose accuracy is an important aspect of validating the effectiveness of our method.
>
> We adopted a dense matching-based evaluation. Specifically, we remove the background from $I_{out}$, resize it to the same resolution as $I_{geo}$, and use MASt3R to estimate dense correspondences on the object regions between the two images. We then compute the average pixel error over the matched points.
>
> |Group|Method|Match Error ↓|
> |---|---|---:|
> |FLUX|Object3DIT + IA|98.9|
> |FLUX|TRELLIS + IA|19.6|
> |FLUX|Ours (FLUX)|**17.8**|
> |SD|Object3DIT + AD|135.7|
> |SD|TRELLIS + AD|75.4|
> |SD|Ours (SD)|**21.4**|
>
> Our method achieves the best overall performance in both the SD-based and FLUX-based categories.
>
> ---
>
> **W3. Inference Latency, Memory Overhead.**
>
> We thank the reviewer for this helpful suggestion. We report inference latency and memory overhead in SD categories. We break down the runtime into the 3D generation stage, the 2D generation stage, and the remaining processing time, and also report the overall end-to-end latency and peak allocated GPU memory.
>
> |Method|3D (s)|2D (s)|Remaining (s)|Overall (s)|Memory (GB)|
> |---|---:|---:|---:|---:|---:|
> |Object3DIT|2.143|6.031|0.436|8.610|9.987|
> |TRELLIS|5.054|6.031|0.412|11.50|11.790|
> |Ours|5.054|4.212|0.276|9.542|10.047|
>
> The overall latency is comparable to baseline models. Our 3D generation stage is slower than Object3DIT, but this cost is incurred upfront before interaction, since the user starts specifying the pose and insertion region after the 3D proxy has been generated. By contrast, our 2D generation stage is faster than the other compared methods. Overall, this leads to competitive end-to-end latency. We also observe that our peak GPU memory usage is comparable to the other methods in this setting.
>
> ---
>
> **W4. Additional ZeroComp Baseline.**
>
> We thank the reviewer for this helpful suggestion.
> Following it, we additionally include both quantitative and qualitative evaluations of a baseline combining TRELLIS and ZeroComp. In this baseline, TRELLIS provides the generated 3D proxy and ZeroComp performs intrinsic-guided compositing for insertion.
>
> |Method|PSNR ↑|SSIM ↑|LPIPS ↓|CLIP ↑|DINO ↑|Match Error ↓|
> |---|---:|---:|---:|---:|---:|---:|
> |ZeroComp|20.66|0.816|0.297|0.892|0.828|5.2|
> |Ours (SD)|21.66|0.829|0.206|0.937|0.913|21.4|
> |Ours (FLUX)|**23.09**|**0.871**|**0.147**|**0.959**|**0.936**|17.8|
>
> As shown in the table, this baseline achieves very low match error, reflecting strong adherence to the rendered proxy. This is expected, since ZeroComp uses intrinsic maps from the 3D proxy as guidance. However, it performs substantially worse on image fidelity and identity preservation, as these conditions do not explicitly preserve the reference object's appearance.
>
> By contrast, our goal is to insert the reference object with both pose control and strong appearance preservation. Under this task requirement, our method performs better overall than ZeroComp.
>
> This difference is also more directly illustrated in the qualitative comparisons shown in Fig. c in the link (https://anonymous.4open.science/api/repo/ICML21834_Rebuttal_CUXV/file/figures.pdf?v=01ebf413).
>
> ---
>
> **Q. Additional Video Explanation to Video Demo**
>
> We thank the reviewer for checking the supplementary video demo and for this helpful suggestion. Briefly, the user provides an object image and a background image, and the object is segmented and lifted into a 3D proxy. The proxy is shown in an interactive interface with a transform gizmo over the background, and the user can drag and rotate it to specify the desired insertion region and pose, after which the system starts generation.
>
> We will also incorporate more detailed audio narration in future versions.

---

> > ### Author Rebuttal · Reviewer_CUXV · 2026-04-01
> >
> > Thank you for the detailed rebuttal. My concerns have been resolved. I will raise my score to support the acceptance of the paper.

---

> > > ### Author Response · Authors · 2026-04-02
> > >
> > > Thank you very much for your follow-up and for updating your score. We are very glad that our rebuttal helped address your concerns. We sincerely appreciate your time, feedback, and consideration.

---

### Official Review · Reviewer_9UnV · 2026-03-13

**Soundness:** 3
**Presentation:** 3
**Significance:** 3
**Originality:** 3
**Overall Recommendation:** 4
**Confidence:** 3

**Summary:**

This paper focuses on the object insertion image editing problem. Existing methods usually treats insertion as a 2D inpainting task, lacking explicit control over the 3D pose (orientation, scale, and placement) of the inserted object, leading to geometric inconsistencies between the object and background. To mitigate the issue, the paper proposes decomposing the insertion condition into three visual proxies: appearance guidance, geometry guidance, and target-integration guidance. Through these decomposed visual proxies, it enables a more intuitive and geometrically accurate workflow for 3D-aware object insertion. In general, the paper explores to leverage interactive 3D manipulation for high-fidelity 2D image editing tasks.

**Compliance With Llm Reviewing Policy:**

Affirmed.

**Final Justification:**

Thanks for the response. I will keep my original score.

**Key Questions For Authors:**

I will adjust my score based on the author's response.

**Limitations:**

Please see the weakness part.

**Strengths And Weaknesses:**

Strength:
1. This paper reformulates the object insertion task as a 3D-aware intervention. Since object insertion inherently involves complex 3D properties (such as orientation, scale, and placement), this approach represents a significant advancement over previous 2D-centric inpainting methods. It is particularly effective for objects with distinct 3D features, ensuring much better geometric alignment with the background.
2. The experimental results are impressive, demonstrating superior visual fidelity and controllability. Furthermore, the paper is well-written; the core concepts and the experimental setup are presented in a clear and easy-to-follow manner.

Weakness:
1. Sensitivity to 3D Proxy: It seems that the framework relies on user-adjusted 3D proxy for geometry guidance. However, the paper lacks a detailed sensitivity analysis regarding the precision of this proxy. It is unclear whether slight misalignments—such as an object being placed slightly above the ground—would be automatically corrected by the proposed approach, or if the model would strictly follow the geometric prompt, leading to unnatural artifacts.

2. Performance on Complex Environmental and Occlusions: I am very curious about the occlusions. For instance, if an inserted object is partially blocked by some elements in the background image, it is unclear how the propose method deal with the depth and lightning conflict. Furthermore, the paper could benefit from a more rigorous discussion on high-frequency light-material interactions, such as how a highly reflective object adapts its appearance to the surrounding environment's lighting. Object insertion in a complex envirionment with high occlusions is very important and challenging. I hope the author could provide a clear analysis.

---

> ### Author Rebuttal · Authors · 2026-03-31
>
> Anonymous figures for this response are provided here: https://anonymous.4open.science/api/repo/ICML21834_Rebuttal_9UnV/file/figures.pdf?v=71f80889
>
> **W1. Sensitivity to 3D Proxy.**
>
> We thank the reviewer for raising this point. We agree that sensitivity to the 3D proxy is an important factor in determining the practical usability of our method.
>
> We conducted an analysis and provide two representative examples in Fig. a in the link:
> - In the first example, the user-adjusted object is placed slightly above the ground.
> - In the second example, the object is not perfectly aligned with the supporting surface.
>
> In both cases, our method is able to automatically correct such mild misalignment and generate natural insertion results. These examples suggest that our method does not require the user to specify a perfectly precise pose, which makes the interaction more user-friendly.
>
> ---
>
> **W2. Performance on Complex Environments.**
>
> We thank the reviewer for raising this point.
> Occlusion, lighting, and reflection are all challenging aspects of object insertion in real scenes.
>
> While our method does not explicitly model physical interactions, illumination, or view-dependent material effects, we find that it can still produce visually plausible results in these challenging cases. We provide three representative examples in Fig. b in the link.
>
> **Occlusion:** We show an example where a pen is inserted into a pen holder. The generated result exhibits a plausible depth relationship between the pen and the holder structure, yielding a natural insertion result.
>
> **Lighting:** We show an example where a car is inserted into a scene with strong directional illumination. The model generates a plausible shadow for the inserted car, improving its consistency with the surrounding scene.
>
> **Reflection:** We note that faithfully modeling reflections on the inserted object itself is not the focus of this work. This is because reflections often alter the object's visible appearance, which can conflict with our main objective of preserving the reference object's identity under user-specified pose control. We therefore leave reflections on the inserted object for future work.
> That said, we find that the method can handle reflective background surfaces reasonably well. As shown by the example of inserting a boat onto water, the generated result includes a visually plausible reflection on the water surface.

---

> > ### Author Rebuttal · Reviewer_9UnV · 2026-04-03
> >
> > Thanks for your response! My concerns are almost addressed. I hope that the author could add these results into the revised version. I will keep my original score.

---

> > > ### Author Response · Authors · 2026-04-05
> > >
> > > Thank you very much for your follow-up. We appreciate your suggestion and will include these additional results in the revised version. We sincerely appreciate your time, feedback, and consideration.

---

### Official Review · Reviewer_kKkY · 2026-03-13

**Soundness:** 3
**Presentation:** 3
**Significance:** 3
**Originality:** 3
**Overall Recommendation:** 4
**Confidence:** 3

**Summary:**

This paper is about inserting 3D objects into 2D images, especially making sure they are aligned with the exact direction the user specified. The authors decompose the insertion conditions into three components: visual guidance from the reference object, geometry guidance from the user-specified 3D proxy, and the target background image.  These condition images are fed into Diffusion Transformer (DiT) blocks to predict and reconstruct the final edited image. Furthermore, the authors propose an automated object curation pipeline to generate the necessary training data. Experimental results demonstrate that the proposed method achieves promising performance when built upon existing generative models such as Stable Diffusion (SD) and FLUX.

**Compliance With Llm Reviewing Policy:**

Affirmed.

**Final Justification:**

The additional experimental results and the explanation have addressed my concerns. I will keep my overall recommendation as "weak accept".

**Key Questions For Authors:**

Please refer to the question I raised in the weakness part.

**Limitations:**

yes.

**Strengths And Weaknesses:**

Strengths:
1. The Automated object curation proposed by the authors is an interesting and scalable pipeline. From my understanding, it would greatly benefit the 2D/3D editing community.
2. The authors conducted thorough experiments, including those based on SD/FLUX and comparisons with methods based on Object3DIT/TRELLIS, demonstrating the effectiveness of the proposed method.
3. The authors' analysis of failure cases and the results from the VAE decoder is very detailed, proving that reconstruction quality and the generated baseline both affect the experimental results.

Weaknesses:
1. Since the training data comes from the pipeline constructed by the authors themselves, are there potential problems? There might be bias or errors in the data construction process. It seems that the authors lack discussion on this aspect.
2. The proposed method can insert objects based on user-specified angles, but it lacks analysis of the rotation angle. For example, what is the maximum possible rotation angle? Are the maximum rotation angles the same for different objects?

---

> ### Author Rebuttal · Authors · 2026-03-31
>
> **W1. Training Data Analysis.**
>
> We thank the reviewer for recognizing the automated object curation pipeline as an interesting and scalable component.
>
> As discussed in the paper (L246--L273), we summarize and further elaborate on these points as follows:
>
> **Strategies to ensure diversity and quality.** Our training set combines automatically curated pairs from SA-1B with filtered multi-view data from MVImgNet. We discuss the strategies to ensure diversity and quality separately:
>
> - **Generated pairs in SA-1B:**
>
>    We impose explicit constraints at each stage of the automatic construction pipeline.
>
>    1. In the object image curation stage, every stage is monitored to retain images containing fully visible, unoccluded objects with precise object masks.
>
>       - In the propose step, a VLM agent identifies salient object categories in the scene and excludes images without suitable foreground objects.
>
>       - In the segment step, SAM-3 produces candidate masks for the proposed instances, and we filter out objects whose masks touch the image boundary or are too small, thereby improving object completeness and visual clarity.
>
>       - In the verify step, the agent performs a zoom-in check on a local crop to further assess both structural completeness and boundary precision, discarding truncated or occluded instances.
>
>    2. In the reference image synthesis stage, we use the mask obtained from the first stage to mask out the background. We then employ an image editing model to synthesize a novel-view reference image, allowing it to focus on generating the object itself. At the same time, the original real image is always used as the supervision target, keeping the training signal anchored to real-world captures.
>
> - **MVImgNet:**
>
>    We use Tenengrad, CLIP-IQA, and Q-Align scores to filter out blurry and low-quality samples before incorporating them into the hybrid dataset.
>
> **Verified Benefit.** Empirically, Table 2 shows that training with the hybrid data improves both image fidelity and identity metrics, suggesting that this strategy improves generalization to diverse real-world scenes.
>
> Despite these efforts, since our data is constructed or filtered from existing source datasets, it may still inherit some of their biases, such as category imbalance and dataset-specific collection bias. We aim to minimize avoidable construction errors through explicit filtering and verification. Some residual errors may still remain due to imperfect masks, synthesis artifacts, or occasional failures in the automatic curation process. We will make this clearer in the revision.
>
> ---
>
> **W2. Rotation Angle Evaluation.**
>
> We thank the reviewer for raising this important question.
>
> **Rotation Angle Analysis.** To study how rotation angle affects performance, we further stratify the benchmark using VLM-annotated approximate relative rotation angles and evaluate our method separately on each bin.
>
> | Rotation Angle Range | 0-45 | 45-90 | 90-135 | 135-180 | Overall |
> |---|---:|---:|---:|---:|---:|
> | Image Fidelity (SSIM) ↑ | 0.864 | 0.881 | 0.864 | 0.877 | 0.871 |
> | Identity (CLIP) ↑ | 0.967 | 0.957 | 0.959 | 0.956 | 0.959 |
> | Pose Accuracy (Match Error) ↓ | 18.1 | 19.4 | 17.8 | 15.7 | 17.8 |
>
> As shown in the table, we do not observe a clear degradation trend as the pose change increases. Image fidelity, identity consistency, and pose accuracy remain stable across all bins.
>
> **Maximum Rotation Angle.** Regarding the maximum supported rotation angle, our method allows the user to specify a full 6D pose through the 3D proxy, and therefore does not impose a theoretical upper bound on the rotation angle itself. In practice, however, the difficulty of large pose changes is object-dependent and may vary with factors such as unseen regions, object complexity, and the quality of the reconstructed 3D proxy.
>
> To demonstrate the method's ability to handle substantial pose changes, we provide four representative examples in Fig. a in the link (https://anonymous.4open.science/api/repo/ICML21834_Rebuttal_kKkY/file/figures.pdf?v=44040ac7):
> - The first two examples show typical large pose changes.
> - The third starts from a top-view input object and generates a side-view result.
> - The fourth applies a 180° rotation and requires a counterfactual generation result.
>
> In all four cases, our method produces good results, suggesting that it can handle large pose changes while maintaining pose consistency and appearance fidelity.

---

> > ### Author Rebuttal · Reviewer_kKkY · 2026-04-04
> >
> > The additional experimental results and the explanation have addressed my concerns.

---

> > > ### Author Response · Authors · 2026-04-05
> > >
> > > Thank you very much for your follow-up. We are very glad that our rebuttal helped address your concerns. We sincerely appreciate your time, feedback, and consideration.

---

### Decision · Program_Chairs · 2026-04-30

**Decision:**

Accept (regular)

**Comment:**

The paper proposed a method for 3D-aware object insertion. All reviewers
appreciate the formulation of the problem and the results of the paper. All
concerns from the reviewers are fully resolved during rebuttal. The reviewers
have reached a consensus to accept the paper, and the AC agrees with the decision
and recommends acceptance.